# Juvenile Exposure to BPA Alters the Estrous Cycle and Differentially Increases Anxiety-like Behavior and Brain Gene Expression in Adult Male and Female Rats

**DOI:** 10.3390/toxics10090513

**Published:** 2022-08-30

**Authors:** Laura Yesenia Castillo, Jorge Ríos-Carrillo, Juan Carlos González-Orozco, Ignacio Camacho-Arroyo, Jean-Pascal Morin, Rossana C. Zepeda, Gabriel Roldán-Roldán

**Affiliations:** 1Behavioral Neurobiology Laboratory, Department of Physiology, Faculty of Medicine, National Autonomous University of Mexico, Mexico City 04510, Mexico; 2Comprehensive Biomedicine and Health Laboratory, Biomedical Research Center, Veracruzana University, Xalapa 91190, Mexico; 3Unidad de Investigación en Reproducción Humana, Instituto Nacional de Perinatología, Facultad de Química, Universidad Nacional Autónoma de México, CDMX, Mexico City 04510, Mexico

**Keywords:** bisphenol A, anxiety, puberty, estrous cycle, behavior

## Abstract

Perinatal exposure to bisphenol A (BPA) in murine models has been reported to affect social behavior and increase anxiety. However, there is little information about the effects of BPA exposure during puberty, a period in which sex hormones influence the maturation and differentiation of the brain. In this work, we evaluated the effect of BPA administration during the juvenile stage (PND 21–50) on anxiety in male and female rats. Newly weaned Wistar rats were treated with BPA (0, 50, or 500 µg/kg/day) for 30 days. To compare the intra- and inter-sex behavioral profiles, rats were evaluated using four different anxiety models: the Open field test (OFT), the Elevated plus maze (EPM), the Light-dark box test (LDBT), and the Defensive burying test (DBT). Males exhibited a clear-cut anxious profile at both doses in all four tests, while no clear behavioral effect of BPA exposure was observed in female rats. The latter showed an altered estrous cycle that initiated earlier in life and had a shorter duration, with the estrous phase predominating. Moreover, the expression of *ESR1*, *ESR2*, *GABRA1*, *GRIN1*, *GR*, *MR*, and *AR* genes increased in the hippocampus and hypothalamus of male rats treated with 50 µg/kg, but not in females. Our results indicate that BPA consistently induces a higher anxiety profile in male than in female rats, as evidenced predominantly by an increase in passive-coping behaviors and changes in brain gene expression, highlighting the importance of sex in peripubertal behavioral toxicology studies.

## 1. Introduction

Bisphenol A (BPA) is a chemical used in the production of polycarbonate, polyacrylates, polyester, epoxy, and phenolic resins [1] employed in the manufacture of domestic, agricultural, and medical products [2,3]. Therefore, human exposure to BPA is ubiquitous and mainly occurs through ingestion, inhalation, and dermal absorption [4]. Due to its structure, BPA binds to intracellular α and β estrogen receptors (ER) [5,6] and androgen receptors (AR) [7], as well as glucocorticoid receptors (GR) [8] and mineralocorticoid receptors (MR) [9], inducing endocrine disruption in the hypothalamic–pituitary–gonad axis [10,11,12] and hypothalamic–pituitary–adrenal axis [8,9]. Consequently, there is a growing concern regarding the permitted BPA concentrations in humans. In the United States, the Food and Drug Administration (FDA) considers that the No Observed Adverse Effect Level (NOAEL) is 50 µg/kg/bw per day [13], while in Europe, the Tolerable Daily Intake (TDI) was reduced from 50 µg/kg/bw to 4 µg/kg/bw per day and further evaluations are still pending in 2021 [14].

BPA has been frequently detected in human urine, blood, amniotic fluid, fetal serum, and breast milk [15,16]. Most studies conducted in humans and rodents have aimed at evaluating the effects of BPA during pregnancy and lactation, showing that perinatal exposure to this xenobiotic may cause reproductive dysfunctions and alters offspring behavior, such as socialization problems, hyperactivity, increased aggressiveness, learning and memory impairment, and changes in anxiety responses [17,18,19,20,21,22]. However, studies on the effect of BPA on anxiety and depression in children have yielded controversial results regarding its association with sex. While some studies have found a positive correlation between mothers exposed to BPA during the perinatal stage and an increase in anxiety symptoms, mainly in boys [23,24,25], other works have observed the opposite, that is, a clear association with higher anxiety scores in girls [26,27]. Likewise, experimental studies using rodents have produced variable results. In general, an anxious phenotype has been observed in rats and mice of both sexes exposed to BPA during gestation and lactation [22,28]. Meanwhile, there are inconsistencies regarding anxiety levels and their possible relationship to sex depending on a variety of factors, including BPA dose, route of administration, exposure window, subject’s age at behavioral assessment, and the anxiety test employed. Few studies have been carried out to determine the effect of rodent exposure to BPA during the post-lactation juvenile stage. In one study, mice were given s.c. BPA (50 µg/kg) daily from PND 23 to 30 and tested as young adults (60–70 PND). BPA-treated males showed less anxiety during testing while an anxiogenic effect was observed in females, using the Open field test (OFT), Elevated plus maze (EPM), New cage test, and Social interaction test [29]. However, the same dose supplied through the diet from PND 35 to 71 increased anxiety in male mice tested in the OFT and Light-dark box test (LDBT) [30]. Likewise, long-term daily intraoral administration of BPA (40 or 400 µg/kg) for 8 weeks (PND 32 to 87) increased anxiety in male mice, but an anxiolytic effect was found in females tested on the OFT and EPM [31]. In rats, an increase in anxiety was observed regardless of sex after daily BPA administration (40 µg/kg) from PND 49 to 60, evaluating anxiety in the OFT and EPM during the last week of treatment [32], though no effect was observed in a setup in which BPA was given from PND 42 to 49 and anxiety was tested in adulthood (PND 77) [33]. Finally, recent research in newly weaned rats treated with BPA (40, 400, and 4000 µg/kg) from PND 21 to 49 and tested in the OFT immediately after the last dose showed a marginal increase in anxiety only in males [34]. Considering the scarce and contradictory information about BPA effects on prepubertal individuals, a period of life in which steroid hormones profoundly influence sex-related brain maturation, we aimed to analyze the effects of BPA administration during the juvenile stage [35,36] on the performance of male and female rats, as adults, in four anxiety tests. In addition, the onset and progression of the estrous cycle were examined. Finally, gene expression of key receptors involved in anxiety response, namely, α-estrogen receptor (*ESR1*), β-estrogen receptor (*ESR2*), α1 subunit of GABAA receptor (*GABRA1*), glutamate receptor subunit 1 (*GRIN1*), glucocorticoid receptor (*GR*), mineralocorticoid receptor (*MR*), and androgen receptor (*AR*) was evaluated in the hippocampus and the hypothalamus.

## 2. Methods

### 2.1. Animals

Newly weaned Wistar male and female rats at postnatal day (PND) 21 obtained from the animal facility of the Faculty of Medicine, National Autonomous University of Mexico, Mexico City, were used. All experimental procedures were performed in accordance with the guidelines and standards established by the Ethics Committee of the Faculty of Medicine, UNAM, Mexico City, project registration number FM/DI/046/2018, and by the Internal Committee for Use and Care of Laboratory Animals No. 006/CIC/2018. Animal care was carried out according to the “International Guiding Principles for Biomedical Research Involving Animals”, Council for International Organizations of Medical Sciences, 2012. Animal Research: Reporting of In vivo Experiments (ARRIVE) guidelines were also followed. Efforts were taken to minimize animals’ suffering throughout all experimental procedures. Animals were maintained in individual Plexiglas boxes in a room with controlled temperature (21 ± 1 °C) and relative humidity (50 ± 10%) under a 12 h light/dark cycle (lights-on at 8:00 h). Glass bottles were used to avoid animal contact with plastic. Animals had free access to water and food and were weighed daily to adjust the dose of BPA they should receive. Six independent groups of rats (three groups of females and three of males, *n* = 8–17/group) were orally administered with vehicle (distilled water), 50 or 500 μg/kg BPA, daily for 30 d, from PND 21 to 50 (Figure 1).

### 2.2. BPA Preparation and Administration

BPA (Sigma-Aldrich, St. Louis, MO, USA, CAS no. 80-05-7, 99% purity) was used. The stock solution contained 50 mg of BPA dissolved in 1 mL absolute ethanol; for the 50 µg/kg treatment, 1.0 µL of stock solution was diluted in 1 mL of distilled H_2_O; while for the 500 µg/kg treatment, 10.0 µL of the stock solution in 1 mL of distilled H_2_O was used; a final volume of 1 mL/kg was orally administered directly into the mouth, through a micropipette. The animals quickly got used to this procedure and accepted the daily administration of BPA-containing water without rejecting it.

### 2.3. Estrous Cycle Evaluation

The phase of the estrous cycle was determined by vaginal smears extracted daily between 10:00–11:00 h, starting from PND 28 until the end of the behavioral tests. Vaginal opening occurred on different days both intra and intergroup and was used to determine the onset of puberty. The four phases of the estrous cycle were established according to vaginal cytology [37] as follows: proestrus (predominance of round and nucleated cells); estrus (predominance of cornified cells); metestrus (presence of all cell types: round and nucleated cells, leukocytes and/or cornified cells); diestrus (predominantly leukocytes) (Appendix A). 

### 2.4. Behavioral Tests

Behavioral tests began 24 h after the last BPA administration. All tests were videotaped, and a blind analysis was performed “off-line” later. Defensive burying test (DBT) recordings were analyzed with Solomon.exe software (www.solomoncoder.com). Recordings of Open-field test (OFT), Elevate-plus maze (EPM), Light-dark box test (LDBT) were analyzed with the Animal Tracker plugins of ImageJ 1.51j8 software. All devices and apparatuses were cleaned with 70% ethanol after testing each rat.

### 2.5. Open-Field Test (OFT)

The OFT arena consisted of a white acrylic square of 100 × 100 × 45 cm placed on a black acrylic sheet of 100 × 100 cm. During testing, light was maintained at 160 lux and rats were allowed to move freely all around the arena for 5 min. This test was used for two purposes: to quantify locomotion and to determine their anxiety level. To assess locomotion, the total distance traveled was recorded, while anxiety was evaluated by the distance traveled and the time spent in the central area of the arena, as previously reported, such that the longer the exploration of the central area of the field the lower the anxiety levels [38].

### 2.6. Elevate-Plus Maze (EPM)

The EPM test was performed in a black acrylic apparatus constructed in accordance with a previous study [39]. The maze consisted of two open (50 × 10 cm) and two enclosed (50 × 10 × 40 cm) arms with an open roof. The arms are intersected at the central square (10 × 10 cm). The maze was elevated 50 cm from the floor by a pedestal fixed under the central square. To avoid rats falling from the maze, acrylic ledges (0.5 × 0.5 cm) were attached along the edges of the open arms. Rats were placed on the central square of the maze facing an open arm at the beginning of the test, and were allowed to explore the maze for 5 min. The number of entries to the open arms (expressed as the percentage of the total number of arm entries), and the total time spent in these arms of the maze were taken as an anxiety index (the higher the time the lower anxiety) [40]. An entry was counted when the four paws of the rat were located in the respective arm. During testing the illumination level was 60 lux.

### 2.7. Light-Dark Box Test (LDBT)

This test was based on the innate aversion of rodents to bright light and their tendency to show exploratory behavior in response to new environments [41]. The LDBT was carried out in a box with two compartments of 30 × 30 × 40 cm each, connected by a 10 × 10 cm opening. The illuminated compartment was made of transparent acrylate walls provided with 92-lux lighting, while the dark compartment had opaque black walls and was poorly (4 lux) illuminated. The test lasted 5 min and was performed as described previously [40]. At the start of the test, each rat was placed in the dark compartment and the latency to the first entry with all four paws into the illuminated compartment was recorded; the time spent in the illuminated and dark compartments, and the number of transitions between the compartments were also recorded. The higher the time spent in the illuminated compartment, the lower the anxiety.

### 2.8. Defensive Burying Test (DBT)

Behavioral testing was performed as described previously [42]. The test was conducted in an acrylic cage (28 × 32 × 45 cm) with the floor covered with a uniform layer (5 cm) of fine sawdust. The cage was equipped with an electrified probe (7 cm long, 0.5 cm thick), which protruded from one of the lateral walls, 5 cm above the bedding. Contact with the probe produced an electric shock of 0.45 mA. The current was generated by a constant current shocker PCAJA-001 (Electronica Steren^®^, Mexico City, Mexico). During the test, two parameters were recorded: the total cumulative time the rat spent burying the probe either with the forepaws or the hind paws (burying behavior), and the total time spent motionless except for tiny and slow lateral movements of the head or those needed for breathing (freezing behavior).

### 2.9. RNA Isolation and Reverse Transcription-Polymerase Chain Reaction 

After behavioral tests, rats were euthanized by decapitation, and the hypothalamus and the hippocampus were dissected and frozen at −70 °C until used. Total RNA was obtained by homogenizing tissues in a Polytron PCU11 homogenizer (Kinematica, Malters, Switzerland) with TRIzol Reagent (Thermo Fisher Scientific, Waltham, MA, USA) according to the manufacturer’s protocol. RNA concentration was measured in the NanoDrop 2000 Spectrophotometer (Thermo Fisher Scientific, Waltham, MA, USA); the integrity of RNA samples was determined in 1.5% agarose gel electrophoresis. One µg of total RNA was employed to synthesize the first-strand cDNA by using the M-MLV reverse transcriptase enzyme (Thermo Scientific, Waltham, MA, USA) following the manufacturer’s instructions. Two μL of synthesized cDNA were subjected to end-point PCR using Vivantis Taq DNA polymerase (Vivantis Technologies, Shah Alam, Malaysia). PCR conditions were as follows: 2 min incubation at 94 °C followed by 30 cycles of 30 s at 94 °C, 30 s at 60 °C, and 30 s at 72 °C, and a final incubation for 7 min at 72 °C. The 18S ribosomal gene was used as an internal expression control. All primers employed in this study are listed in Appendix A. PCR products were observed in 1.5% agarose gels; relative expression level was obtained for each gene by densitometric analysis. At least three biological replicates for each experiment were carried out.

### 2.10. Statistical Analysis

All data were plotted and analyzed with the GraphPad Prism 7^®^ software (GraphPad, San Diego, CA, USA). When the experimenter performing the blind offline analysis suspected the presence of an outlier, a Grubb test was performed, and outliers with a significance level of *p* < 0.01 were removed from the analysis. Behavioral data were analyzed using a two-way analysis of variance (ANOVA). Estrous cycle data were analyzed using a one-way ANOVA. Tukey post-hoc tests were conducted where applicable. Gene expression analysis was carried out using One-Way ANOVAs with post-hoc Tukey tests. Values of *p* < 0.05 were considered statistically significant.

## 3. Results

### 3.1. Body Weight

Male and female rats of the three groups (vehicle, BPA 50 µg/kg and BPA 500 µg/kg) were weighed daily from PND 28 to PND 54 during the first set of experiments (*n* = 9/group). No significant weight × treatment interaction was observed in male rats (F_(34,408)_ = 0.61) nor in female rats (F_(34,408)_ = 1.37) (Appendix A). A slight main effect of treatment was observed in females only (F_(2,24)_ =3.56, *p* < 0.05), although post-hoc Tukey tests unveiled no significant differences among groups (*p* > 0.07).

### 3.2. Estrous Cycle 

The estrous cycle was monitored daily, from PND 28 until the end of the behavioral tests (PND 54). The phase of the cycle in which females were on the vaginal opening day did not always correspond to proestrus, as has been previously reported [43,44]. Premature onset of puberty (vaginal opening day) in the BPA-treated rats compared to the vehicle-treated group was observed (PND = 30.89 ± 0.35, 29.11 ± 0.30 and 29.00 ± 0.19, for vehicle, BPA 50 and BPA 500 µg/kg, respectively, F_(2,32)_ = 14.25, *p* < 0.0001), (Figure 2a). BPA treatment at 50 and 500 µg/kg also increased the number of cycles (as estimated by the number of estrous) from 5.11 ± 0.6 to 8.89 ± 1.4 and to 8.77 ± 0.8, respectively, (F_(2,32)_ = 50.74, *p* < 0.001, Figure 2b) and decreased estrous cycle length, measured as the interval between estrous and estrous in days, from 3.96 ± 0.1 to 2.66 ± 0.1 and 2.67 ± 0.1, respectively (F_(2,32)_ = 68.09, *p* < 0.0001, Figure 2c). Moreover, we observed changes in proestrus, estrus, and metestrus frequency where a main effect of treatment (F_(2,32)_ = 19.17, *p* < 0.0001) and a significant cycle stage X treatment interaction (F_(6,96)_ = 19.59, *p* < 0.0001) were observed (Figure 2d). Additionally, a shorter proestrus (*p* < 0.001) and a longer estrus (*p* < 0.0001) in 50 or 500 µg/kg BPA-treated rats were detected compared to the vehicle group. Regarding metestrus, both 50 µg/kg and 500 µg/kg BPA-treated groups showed a shorter period than the vehicle group (*p* < 0.05 and *p* < 0.01, respectively), while the diestrus phase did not change (*p* > 0.11). Noteworthy, as discussed below, we did not find any effect of the estrous phase on behavior in the anxiety tasks detailed ahead (see also Appendix A).

### 3.3. Anxiety-like Behavior

#### 3.3.1. Open Field Test (OFT)

The effects of BPA on locomotion and anxiety as evaluated in the OFT are shown in Figure 3. Panel A shows representative trajectories for male and female rats of the three groups. Overall, the total distance traveled by females was greater than males, regardless of treatment (F_(1,64)_ = 16.84, *p* < 0.001), suggesting that although females tended to explore more, BPA treatment and either dose did not produce a locomotor impairment, nor did it affect the rats’ motivational state (Figure 3b). Meanwhile, the distance traveled in the center of the arena decreased in subjects treated with BPA compared to the vehicle group (F_(1,62)_ = 4.34, *p* < 0.05); no effect of sex (F_(2,62)_ = 0.65, *p* = 0.42) or interaction (F_(2,62)_ = 2.24, *p* = 0.12) was observed. Post-hoc Tukey analysis unveiled a significant effect in the 500-µg group only (*p* < 0.05, Figure 3c). Likewise, BPA treatment significantly decreased the time spent in the central zone (F_(2,64)_ = 6.15, *p* < 0.01) regardless of sex (F_(1,64)_ = 0.01, *p* = 0.93); no sex-treatment interaction was observed (F_(2,64)_ = 1.07, *p* = 0.35 (Figure 3d). Again, post-hoc Tukey analysis unveiled that only the rats treated with the 500-µg dose differed significantly from controls (*p* < 0.01).

#### 3.3.2. Elevated Plus Maze (EPM)

The EPM test takes advantage of the natural aversion of rats towards open and high areas. In this test, it is assumed that the higher the number of entrances and the time spent in the open arms of the maze, the lower the anxiety level. Figure 4 depicts the effect of PBA on EPM performance. Panel A shows representative trajectories for male and female rats of the three groups. First, even more evident than in the OFT, statistical analysis showed a robust main effect of sex as females tended to spend much more time exploring the open arms, irrespective of treatment (F_(1,61)_ = 31.14, *p* < 0.0001, Figure 4). BPA treatment was also found to significantly affect the time spent in the open arms (F_(2,61)_ = 14.03, *p* < 0.0001); males treated with the 50 and 500 µg doses explored significantly less than their vehicle counterparts (*p* < 0.01 and *p* < 0.05, respectively) as occurred with females in both treatment groups (*p* < 0.01). Similarly, the percentage of entries was higher in females than in males (F_(1,60)_ = 7.23, *p* < 0.01) and a robust effect of BPA treatment was observed (F_(2,60)_ = 12.10, *p* < 0.0001). Specifically, males treated with both 50 and 500 µg doses showed a decreased percentage of entries in open arms (ps < 0.001 and 0.01, for the 50 and 500 µg groups, respectively) although, interestingly, no significant effect of BPA treatment was observed when comparing females treated at either dosage with their vehicle counterparts (ps > 0.11). These observations are consistent with an anxiogenic effect of BPA that is more pronounced in male subjects.

#### 3.3.3. Light-Dark Box Test (LDBT)

The effects of BPA on LDBT performance are shown in Figure 5. Panel A illustrates representative trajectories for male and female rats of the three groups. Statistical analysis revealed significant effects of sex (F_(1,58)_ = 9.47, *p* < 0.01), treatment (F_(2,58)_ = 5.48, *p* < 0.01), and sex × treatment interaction (F_(2,58)_ = 9.84, *p* < 0.001, Figure 5) for the time spent in the lit box. Importantly, post-hoc analysis revealed a significant effect of BPA treatment in males in the 500-µg group only (*p* < 0.0001), which spent significantly less time in the lit compartment, while the xenobiotic had no effects in females (ps > 0.95). These results suggest that peri-adolescent BPA treatment increased photophobia in males, only at the highest dose used. We next compared the latency to enter the lit box among the distinct BPA groups for each sex. Here, no significant effect of sex or treatment was observed, nor did we find any interaction (*p* > 0.34 in all cases). Finally, the comparison of the number of entries to the lit box between each group revealed a significant effect of sex (F_(1,58)_ = 29.75, *p* < 0.0001) and treatment (F_(2,58)_ = 4.55, *p* < 0.05) but no sex × treatment interaction (F_(2,58)_ = 2.04, *p* > 0.13); females entered the lit box significantly more times than their male counterparts. 

#### 3.3.4. Defensive Burying Test (DBT)

The previous experiments aimed to evaluate the response to different stressors in male and female rats exposed to BPA during puberty. These tasks, however, do not distinguish between distinct types of stress-induced behaviors. The DBT has been argued to do just that, distinguishing between active and adaptive (burying) and passive and maladaptive (freezing) behaviors induced by stress [45]. When evaluating the time spent burying, statistical analysis unveiled significant main effects of both sex and treatment (F_(1,53)_ = 20.42, *p* < 0.001 and F_(2,53)_ = 5.02, *p* < 0.05 for sex and treatment, respectively) as well as a significant interaction (F_(2,53)_ = 5.14, *p* < 0.01). Overall, females spent burying more time than males, although they did not differ between treatment groups (*p* > 0.18, Figure 6A). In male rats, post-hoc tests showed that animals with 500 µg/kg treatment buried significantly less than those in the 50 µg/kg group although both groups did not differ from the vehicle (*p* > 0.38). On the other hand, a profound dose-dependent increase in the time spent freezing was observed in male rats, in which those treated with the 500 µg/kg dose spent significantly more time freezing than the control and 50 µg groups (*p* < 0.0001 in both cases) (Figure 6B). As a proportional measure of the effect of BPA on passive *versus* active stress coping strategies in rats of both sexes, the bury time ratio was calculated as (time spent burying)/(time spent burying + time freezing). Significant effects of sex (F_(1,46)_ = 40.14, *p* < 0.0001) treatment (F_(2,46)_ = 23.24, *p* < 0.0001) and sex × treatment interaction (F_(2,46)_ = 15.03, *p* < 0.0001) were observed. Post-hoc analysis showed that this ratio was similar in females regardless of BPA exposure. In contrast, in males it was significantly lower in the 500 µg/kg group (*p* < 0.0001), suggesting that in these animals, exposure to the higher dose of the xenobiotic produced a shift in the coping strategy that favored a passive response to stressful situations. 

### 3.4. Juvenile BPA Exposure Alters Gene Expression in the Hippocampus and the Hypothalamus

Next, we sought to evaluate the mRNA levels of genes associated with anxiety responses in the hippocampus and hypothalamus of BPA-exposed male and female rats. In the hippocampus of males, we detected significant differences in mRNA levels among treatment groups in all the studied genes (*ESR1*, *ESR2*, *GABRA1*, *GRIN1*, *GR*, *MR*, and *AR*) (One-way ANOVAs Fs ≥ 4.95, *p* < 0.05). The group receiving 50 μg/kg of BPA expressed higher levels in all analyzed genes (Tukey’s post-hoc test, *p* < 0.05), while the 500 μg/kg BPA group presented higher expression levels of *GRIN1*, *GR*, *MR*, and *AR* than the control group (*p* < 0.05 in all cases). No changes were observed in the mRNA levels in the hippocampus of BPA-treated females (Appendix A). In the hypothalamus of BPA-treated males, the expression of all studied genes but *ESR1* was significantly different among experimental groups (Fs ≥ 5.88, *p* < 0.05). In the 50 µg/kg BPA group a significant increase was observed in all genes except *ESR1* (*p* < 0.05), while in the 500 µg/kg only *ESR2* and *GR* were increased (*p* < 0.001 and *p* < 0.01, respectively) compared with the control group (Appendix A). Here again, no significant changes were observed in the expression of the tested genes in the hypothalamus of females in any of the BPA concentrations employed (Appendix A). 

## 4. Discussion

In the present work, the effect of exposure to BPA during the juvenile stage, a critical period for hormone-dependent sexual and neurological maturation, on anxiety-like behavior in male and female rats was evaluated. First, we show that juvenile BPA exposure induced noticeable alterations in the estrus cycle; the puberal onset was advanced, the number of cycles increased, decreasing their duration, and the frequency of estrus augmented at the expense of proestrus, which diminished (Figure 2). Our main finding, however, was a sex-dependent anxiogenic profile in adult rats kept under BPA treatment while juvenile, which adds to the neurobehavioral alterations reported in prior studies [22,28]. In all four behavioral tests used, male rats exhibited a significant and reliable increase in anxiety-like responses, while females only showed an anxiogenic profile in the EPM test. Finally, our results from the DBT test suggest that BPA exposure during the juvenile stage may promote a shift from active to passive-coping behavior in males during anxiety-prompting events, a behavioral alteration that has been observed in preclinical models of depression and PTSD [46,47]. 

Considering that BPA is an endocrine disruptor [5,6], it was essential to determine how this compound affects the estrous cycle, as it has long been claimed that rodent behavioral performance in anxiety tests varies throughout its different phases [48,49]. Moreover, there were no studies evaluating the influence of the estrous cycle phase on anxiety-like behavior in subjects exposed to BPA. In our study, BPA-treated females were mainly tested in the estrus and diestrus phases due to its higher prevalence during the testing sessions, but we did not observe any significant difference from those females evaluated in proestrus or metestrus. Hence, we did not find any correlation between the phase of the estrous cycle and anxiety levels in BPA-treated rats (Appendix A), in contrast to previous reports in intact, spontaneously cycling rodents [48,49,50]. This difference might be due to the complexity of disrupting actions caused by BPA on diverse targets, including the reduction in ovarian steroidogenic activity together with modifications of LH or GnRH release and direct agonism of estrogen receptors in the central nervous system, which could mask the hormonal influence on anxiety observed in naive animals [12,51]. Curiously, our control rats did not show a cycle phase-dependent behavioral performance either (see Appendix A), which agrees with recent studies that found no significant correlation between anxiety and the estrous cycle phase [52,53]. 

Our findings of an earlier vaginal opening in rats exposed to BPA are in line with prior investigations in which prenatal [54] and postnatal BPA treatment accelerates the onset of puberty in rats [12,55] and mice [56]. Here we show that a brief (from PND21 to PND28) post-weaning exposure to this endocrine disruptor was enough to induce an advance of puberty onset. Moreover, as the estrous cycle was monitored for a relatively long period, we were able to verify that the vehicle-treated rats showed a regular cycle from the third estrus, while those receiving BPA never consolidated a regular cycle throughout its progression. 

Regarding anxiety changes after BPA treatment, research in animals and humans has shown, as stated in the Section 1, that exposure to this xenobiotic during critical periods of development, primarily pregnancy and lactation, has a detrimental outcome on brain functions manifested as altered behavioral expressions including hyperactivity, aggressiveness, memory decline, and increased anxiety [17,22,24,57,58]. However, exposure to BPA is ubiquitous and continuous throughout life due to the diversity and availability of potential contamination sources. Therefore, examining its effects during childhood and adolescence is of great interest but has been scarcely studied. Besides, puberty is a vulnerable developmental period and the existing reports on juvenile exposure to BPA are contradictory [31,33,34,59]. In this regard, it is well known that both anxiogenic and anxiolytic effects of a variety of treatments are not always manifested in all behavioral tests designed to assess anxiety preclinically. In the present study, we used four anxiety tests that are based on different conflict situations for rodents, from which OFT and EPM have been widely used to evaluate the actions of BPA, but LDB and DBT have not, at least in rats. Our results were consistent, showing a clear anxiogenic profile in male rats and a less intense and test-dependent one in females, in which we observed an anxiety-promoting effect of BPA in the EPM task only (Figure 4b). Specifically, we observed that 50 and 500 µg/kg BPA-treated male, but not female rats presented an increased anxiety-like behavior in the OFT, avoiding exploration of the central area of the arena without any change in total locomotion, which coincides with some earlier works [29,32,34] but differs from others, in which the authors found anxiolysis or lack of effect [31,33,59]. The main differences between these studies appear to be the exposure time window and the age at which the animals were tested: the younger they were exposed and tested, the greater the effect. On the other hand, in the EPM, our data show that both male and female rats exhibited an anxiogenic profile, agreeing with earlier reports in which rodents were exposed to BPA at various periods of the juvenile stage and assessed as adults. In these studies, male [30,31] and female mice [29] as well as rats of both sexes [32] were rated with high level of anxiety in this test. Moreover, paternal, gestational, and/or perinatal exposure to this xenobiotic elicits marked anxiety-like behavior in female rats and mice [57,58,60]. In the DLBT, only males exhibited photophobia-induced anxiety behavior, similar to what was reported in the offspring of pregnant and lactating mice exposed to BPA (0.4 and 4 mg/kg) when tested as young adults (PND 56) which produced increased anxiety both in males and ovariectomized females [61]. Finally, in the DBT, male rats treated with 50 µg/kg and females treated with 50 and 500 µg/kg showed an increase in burying, an active-cope anxiety behavior, while only males treated with 500 µg/kg showed a rise in freezing, a passive-cope behavior. Since freezing is related to an intense, arguably maladaptive expression of anxiety, we infer that males were again more sensitive to the BPA treatment than females. Indeed, an increase in passive-coping strategy in this test has been suggested to represent a maladaptive response and has been observed in preclinical models of anxiety disorders and depression and has been linked to an increased risk of metabolic and hormonal imbalance as adults [47,62].

We evaluated the expression of several genes participating in activating and regulating the rapid response circuit to stress and anxiety in two regions highly involved in these behaviors: the hippocampus and the hypothalamus [63,64]. The expression of *ESR1*, *ESR2*, *GABRA1*, *GRIN1*, *GR*, *MR*, and *AR* genes was upregulated in the hippocampus and hypothalamus of males treated with BPA but not in females. These results suggest that gene expression in the brains of males is altered due to juvenile exposure to BPA, which may be linked to the anxiety increase responses observed in the behavioral tests.

Few studies have evaluated the effects of BPA exposure during the childhood/juvenile stage at the molecular level, most of them focusing on behavioral effects. A recent report [34] showed that BPA administration from PND 21 to 49 was associated with decreased levels of NR2 receptor subunit in the hippocampus and V1 cortex of male rats as well as decreased ERβ levels in the hypothalamus. The apparent discrepancy with our results could be explained by the fact that in that study, the authors used considerably higher doses (up to 4 mg/kg/day) and that they evaluated the protein rather than the transcript.

Studies evaluating the effect of BPA exposure during the pregnant/lactating stage on anxiety-related gene expression yielded somewhat diverging results. One study reported a decrease in the *ERα* transcript in the hippocampus of male and female rats [65], while another found a decrease in the expression of *ERα* without modifications in *ERβ* in the hippocampus [66]. On the other hand, no modifications of *ER* and *AR* expression were found in the hypothalamus of zebrafish exposed during pregnancy [67]. Interestingly, another group found that gestational exposure to BPA increased expression of *GR* in the hypothalamus and hippocampus of female rats and a decrease in males; in addition, melanocorticotropic hormone receptor mRNA levels were increased in the hippocampus of female rats, while decreased in males [68]. Finally, Xu et al. (2012) reported an increase in the expression of the NMDA receptor in the amygdala and a decrease in the hippocampus [61]. 

In summary, our results indicate that BPA advances the onset of puberty and decreases the estrous cycle duration, predominantly maintaining the estrous stage. Moreover, BPA induces a clear-cut anxiogenic profile in male rats, while females showed a moderate and therefore, test-dependent anxious behavior, which was only observable in the EPM. These changes did not follow a dose-dependent relationship since animals treated with 50 µg/kg were more affected. These behavioral effects were accompanied by an increase in the expression of several genes involved in anxiety in the hippocampus and the hypothalamus of male animals, suggesting a sexual dimorphism in BPA responses in juvenile animals. Our results, especially in the DBT task, suggest that juvenile exposure to BPA exacerbates passive coping and maladaptive anxiety-induced behavioral responses in male rats and supports the notion that this compound represents a risk factor for emotional disorders.

## Figures and Tables

**Figure 1 toxics-10-00513-f001:**
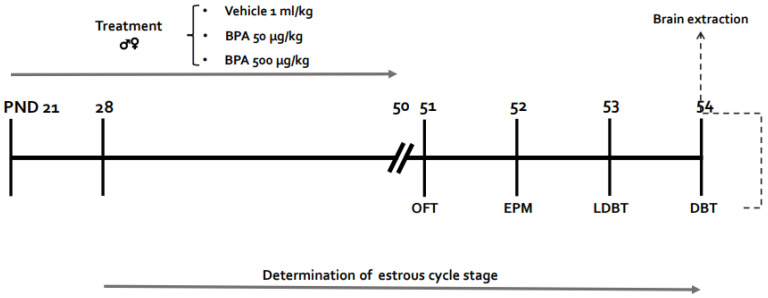
Experimental design. Timeline representing the experimental design followed in this study. OFT: Open field test; EPM: Elevated plus maze; LDBT: Light dark box test; DBT: Defensive burying test; PND: Postnatal day.

**Figure 2 toxics-10-00513-f002:**
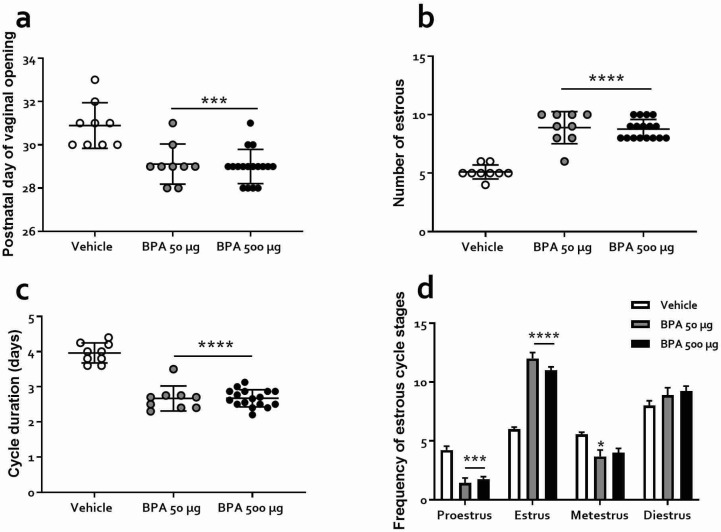
Effects of BPA on the estrous cycle of rats exposed during PND 28–54. (**a**) Onset of the estrous cycle (PND). (**b**) Number of cycles during the PND 28–54 period. (**c**) Length in days of the estrous cycle during the same period. (**d**) Frequency of each stage of estrous cycle. Results are represented as mean ± SEM (*n* = 9). * *p* < 0.05; ***: *p* < 0.001; ****: *p* < 0.0001 vs. vehicle.

**Figure 3 toxics-10-00513-f003:**
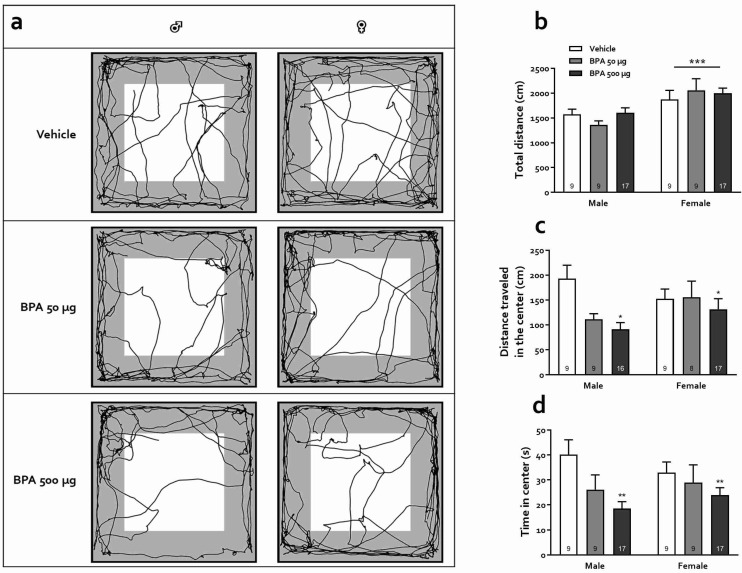
Effects of BPA on the OFT. (**a**) Representative trajectory for each treatment; the gray area corresponds to the periphery and the white area to the center of the arena. (**b**) Time spent in the center of the arena in male and female rats. (**c**) Distance traveled in the center of arena, and (**d**) Total distance traveled. Each bar represents the mean ± SEM. Numbers within bars indicate animals per group. *: *p* < 0.05 and **: *p* < 0.01 vs. pooled vehicle; ***: *p* < 0.001 vs. males.

**Figure 4 toxics-10-00513-f004:**
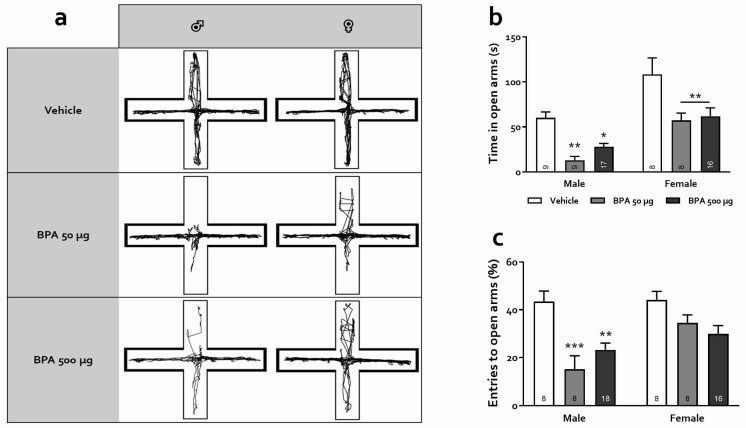
Effects of BPA on the EPM test. (**a**) Representative trajectory for each treatment; thicker contouring lines represent the enclosed arms; (**b**) Time spent in the open arms, and (**c**) Percentage of open arm entries over the total number of open and closed arms entries in male and female rats. Results are expressed as mean ± SEM. Numbers within bars indicate animals per group. *: *p* < 0.05; **: *p* < 0.01; ***: *p* < 0.001 vs. respective vehicle groups.

**Figure 5 toxics-10-00513-f005:**
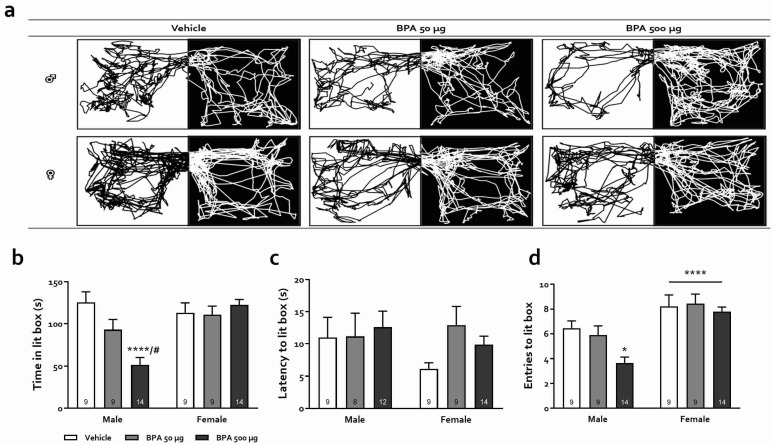
Effects of BPA on LDBT. (**a**) Representative trajectory for each treatment. The black area corresponds to the dark box and the white one to the light box; (**b**) Time spent in the light box for male and female rats; (**c**) Latency to enter the light box. (**d**) Number of transitions to the light box. Results are expressed as mean ± SEM. Numbers within bars indicate animals per group. ****: *p* < 0.0001 vs. male vehicle group; #: *p* < 0.05 vs. male 50 µg group; *: *p* < 0.05 vs. male vehicle group; ****: *p* < 0.0001 male vs. females.

**Figure 6 toxics-10-00513-f006:**
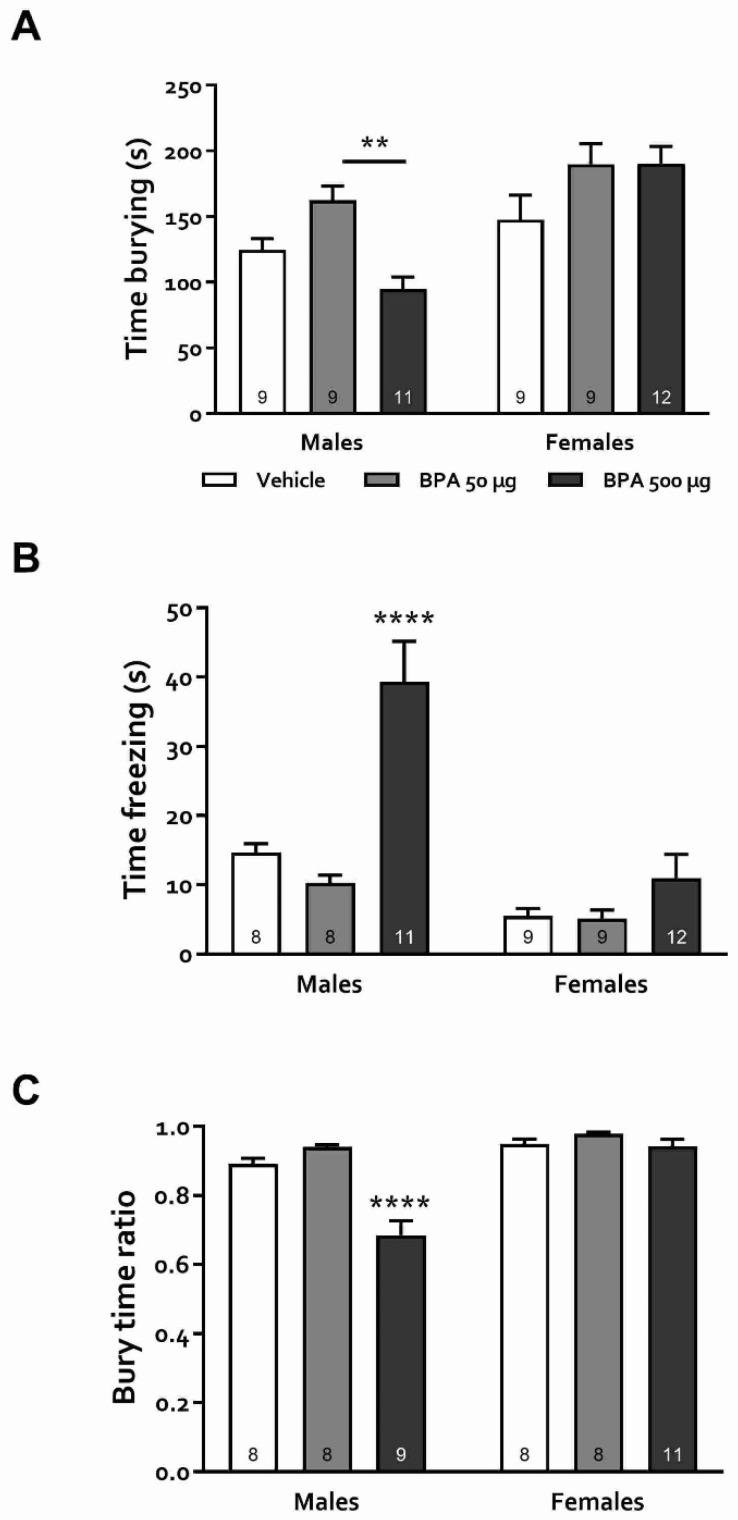
Effects of BPA on active and passive-coping behaviors in DBT. (**A**) Cumulative time spent on burying behaviors. (**B**) Time spent on freezing behavior. (**C**) Bury time ratio: proportion of time spent burying vs. total time spent burying + freezing. Results are shown as mean ± SEM. Numbers within bars indicate animals per group. **: *p* < 0.01; ****: *p* < 0.0001 vs. male vehicle group.

## Data Availability

Data available upon request.

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
