# Peer review of "Juvenile Exposure to BPA Alters the Estrous Cycle and Differentially Increases Anxiety-like Behavior and Brain Gene Expression in Adult Male and Female Rats"

_toxics, 2022, doi:10.3390/toxics10090513_

Round 1

Reviewer 1 Report

The authors investigated the effect of bisphenol A (BPA) on social behavior and anxiety using a rat model. Juvenile female and male rats were treated with BPA (), 50 or 500 µg/kg/day) for 30 days. Afterward, the monitoring of body mass and estrous cycle, and several behavioral tests were performed. Authors found that BPA treatment effect of the estrous cycle in females, while did not affect their behavior. Unlike, BPA induced a higher anxiety profile in males. The results are interesting. There are a few queries that should be addressed.

1.      In addition to study the effect of BPA on female maturation, anogenital distance in males can be measured to study the effect of BPA on male maturation.

2.      Though the authors mentioned the effect of BPA on LH and GnRH release in the discussion, it would be desirable to measure levels of several hormones during BPA treatment (e.g. estrogens, testosterone, LH, FSH). 

3.      The authors noticed the effect of PBA on the estrous cycle. I think that authors could discuss how this effect can contribute to female reproductive health.

Reviewer 2 Report

Bisphenol A (BPA) is a chemical used in the fabrication of various industrial products, including polyester. There has been controversy surrounding BPA exposure, particularly its effects on the endocrine system. In this manuscript, Castillo et al. examined the behavior of puberty rats for 30 days after exposure to 50 or 500 ug/kg BPA. There was a difference between the results of males and females. Estrous cycles in female rats were altered, with early estrus and a greater number of cycles. There were no clear effects on female behavior, but male rats consistently showed increased anxiety in the open field test (OFT), elevated plus maze test (EPM), light/dark box test (LDBT), and defensive burying test (DBT). The study is well designed, and the conclusions are based on consistent results. Many papers have been published regarding the effects of BPA on humans and animals, some of which indicate differences between males and females. Therefore, the novelty of this study cannot be considered very high. However, some of the previous studies contradict each other, and no clear conclusions have been drawn yet. In this reviewer's opinion, this manuscript's novelty does not diminish its value, since the accumulation of multiple studies from different groups will enable us to accurately assess the effects of BPA. Consequently, this reviewer recommends publishing the manuscript with minor revisions.

Minor points

  1. Figure 3d: There is no description of the results of Figure 3d in the main text.

  2. Figure 3: There is no result with *P<0.05 that is indicated in the figure legend.

  3. Figure 5d: It is confusing to present the p-values this way, as if there was a significant difference between all the male and female experiment groups.

  4. Figure 5: Evaluating the results of behavioral experiments requires the correct sample size. Hence, the number of samples per group should be specified as n=9 rather than a range such as n=8-17. For groups with different numbers, this reviewer suggests to include them in the figure if they are difficult to write down.

Round 2

Reviewer 1 Report

I would appreciate the authors comments. The results of analyses of several gene expression, even done by end-point PCR, are interesting and can somehow explain observed changes in male behavior following BPA treatment. I recommend to include these results (it may be in the main manuscript or in supplementary data) and some discussion of possible correlation changes in expression of these genes and their possible contribution to male behavior. I advise to be careful in the writing of figure description. In my opinion, the title  as in revision Fig.1"BPA effects on gene expression associated with the anxiety response in the male and female hippocampus" is not correct because e.g. GABRA-1encode neutrotransmitter that is not directly associated with anxiety, ESR1 encode estrogen receptor that orchestrated various cellular responses and so on.

Author Response

Thank you very much for the comments. As suggested by the reviewer, we now include our data obtained by end-point RT-PCR as supplementary material and discuss the results we obtained in the Discussion section of the main text. We also modified the figure descriptions as recommended. The added text corresponding to our gene expression data appears highlighted in yellow in the newest version of the manuscript.